# Potential Utilization of Bacterial Consortium of Symbionts Marine Sponges in Removing Polyaromatic Hydrocarbons and Heavy Metals, Review

**DOI:** 10.3390/biology12010086

**Published:** 2023-01-05

**Authors:** Ismail Marzuki, Rosmiati Rosmiati, Akhmad Mustafa, Sahabuddin Sahabuddin, Tarunamulia Tarunamulia, Endang Susianingsih, Erfan Andi Hendrajat, Andi Sahrijanna, Muslimin Muslimin, Erna Ratnawati, Kamariah Kamariah, Khairun Nisaa, Susila Herlambang, Sri Gunawan, Idum Satia Santi, Bambang Heri Isnawan, Ernawati Syahruddin Kaseng, Early Septiningsih, Ruzkiah Asaf, Admi Athirah, Basri Basri

**Affiliations:** 1Department of Chemical Engineering, Fajar University, Makassar 90231, South Sulawesi, Indonesia; 2Research Center for Fishery National Research and Innovation Agency, Cibinong 16911, West Java, Indonesia; 3Soil Science Departement of Agriculture Faculty Universitas Pembangunan Nasional Veteran, Yogyakarta 55283, DI Yogyakarta, Indonesia; 4Department of Agrotechnology, Institut Pertanian Stiper, Yogyakarta 55283, DI Yogyakarta, Indonesia; 5Department of Agrotechnology, Universitas Muhammadiyah Yogyakarta, Bantul 55183, DI Yogyakarta, Indonesia; 6Agricultural Technology Education Department, Faculty of Engineering, Makassar State University, Makassar 90222, South Sulawesi, Indonesia; 7Research Center for Conservation of Marine and Inland Water Resources, National Research and Innovation Agency, Cibinong 16911, West Java, Indonesia; 8Institute of Health Science (STIK), Makassar 90231, South Sulawesi, Indonesia

**Keywords:** removal, PAHs, heavy metals, marine sponges, bacterial consortium

## Abstract

**Simple Summary:**

Marine waters are the most susceptible to exposure to contaminants, specifically polyaromatic hydrocarbons (PAHs) and heavy metals that are poisonous and carcinogenic. On the other hand, a marine environment can provide various natural materials that can perform biodegradation and biosorption. This research seeks to investigate various species of symbiotic sponge bacteria and assess the feasibility of developing bacterial consortium formulations for the remediation of these hazardous contaminants. Several species of symbiotic sponge bacteria are capable of biodegrading polyaromatic compounds. According to the search and analysis results, several species of symbiotic sponge bacteria have the potential to biodegrade polyaromatic compounds, such as naphthalene, anthracene, and pyrene. Other sponge symbiont bacteria or bacteria isolated from the same sponge can also be used for the biosorption of heavy metals. The ability of the symbiotic sponge bacteria to be formulated as a consortium of bacteria for pollutant remediation applications, which we refer to as the metallohydrocarbonoclastic formula, is another feature that sets them apart. The proposed use is a bioremediation approach that the community can implement to the isolation processes, such as the remediation of contaminants in water used for fish farming and shrimp plotting methods, including in ponds, to provide aquaculture products free of toxic pollutants.

**Abstract:**

Toxic materials in waste generally contain several components of the global trending pollutant category, especially PAHs and heavy metals. Bioremediation technology for waste management that utilizes microorganisms (bacteria) has not been fully capable of breaking down these toxic materials into simple and environmentally friendly chemical products. This review paper examines the potential application of a consortium of marine sponge symbionts with high performance and efficiency in removing PAHs and heavy metal contaminants. The method was carried out through a review of several related research articles by the author and published by other researchers. The results of the study conclude that the development of global trending pollutant (GTP) bioremediation technology could be carried out to increase the efficiency of remediation. Several types of marine sponge symbiont bacteria, hydrocarbonoclastic (R-1), metalloclastic (R-2), and metallo-hydro-carbonoclastic (R-3), have the potential to be applied to improve waste removal performance. A consortium of crystalline bacterial preparations is required to mobilize into GTP-exposed sites rapidly. Bacterial symbionts of marine sponges can be traced mainly to sea sponges, whose body surface is covered with mucus.

## 1. Introduction

Global trending pollutant (GTP) is a term applied to several types of pollutant materials (heavy metals, aromatic hydrocarbons, microplastics, medical waste, and pesticide residues) that pose many complex problems to the global environment [1,2,3,4,5,6,7]. The problem of environmental quality is especially felt by developing countries [3,8]. The issue of GTP in this decade has been very much discussed, not only by environmental observers and activists, as well as scientists in the field of environmental management, but also by several world leaders who have voiced the need for real action in reducing fossil fuel consumption [9,10,11]. The global policy of reducing carbon consumption is a tangible manifestation of the emergency of environmental quality [7,12]. It is very reasonable because the rate of increase in global trends of pollutants is increasing massively, far beyond the natural recovery ability of nature to reduce these pollutants [13,14,15]. Reducing carbon in the environment is not enough because various types of GTP and their adverse environmental effects are also different [7,16].

The aquatic environment is a giant container that is most vulnerable to being affected by GTP contaminants [17]. This is because the topography of the area is generally low. Almost all types of GTP contaminants are found in particulates and residues and can be dissolved in air, soil and water bodies [4,7]. These materials eventually empty into the aquatic environments of rivers, lakes, swamps and the sea [18,19]. In this environment, these toxic pollutants form parasitic accumulations in almost all marine organisms, especially fish, sponges, algae, plankton, phytoplankton and other types of biota [20,21].

At the same time, much research has been carried out related to the management of toxic pollutants in order to achieve the dream of creating a green environment [22,23]. The results of this research have yielded many findings, technologies, innovations and methods related to removing toxic contaminants [24,25]. Degradation, reduction, destruction, and absorption are recommended and can be applied to decompose carcinogenic pollutants in the environment [26,27,28]. This method can be called bioremediation technology if it involves the role of living organisms, including the contribution of microorganisms as biodegraders that can decompose or may be able to eliminate the toxic properties of GTP contaminants [29,30,31].

The principle of bioremediation technology involves the development of biological methods that can be applied in a gradual combination with physical, chemical, and biological methods, depending on the characteristics of the pollutant being degraded [32,33]. Bioremediation technology that uses microorganisms as a decomposer component is generally good and is more often applied to wastewater treatment [34,35]. Sources of microorganisms that can carry out bioremediation functions can be obtained in an aquatic environment contaminated with pollutants, for example, in port areas, offshore oil processing industries or marine areas around locations that have experienced oil spills [36,37,38]. Pollutant-degrading microorganisms can also be found in the soil, especially on land with a history of contamination due to oil spills and agricultural land exposed to pesticides [3,39]. Toxic pollutant-degrading microorganisms can also be found in other organisms that form a symbiotic relationship, for example, in sponges and mangroves [40,41].

Isolation and screening methods can be applied to obtain potential microorganisms as biomaterials for degrading pollutants [42,43]. In general, two microorganisms have potential as biomaterials for degrading pollutants, namely, bacteria and fungi [44,45,46]. These groups of microorganisms have different abilities and degradation mechanisms for pollutants [47]. The application of bacteria in bioremediation mainly removes hydrocarbon contaminants, especially polyaromatic hydrocarbons (PAHs) and heavy metal pollutants. At the same time, the use of fungi in pollutant bioremediation has also been carried out but is still limited or is not as popular as using bacteria [48,49,50].

The types of bacteria identified have bioremediation capabilities against PAHs, but the level of bioremediation achieved is still low, especially if only one type of bacteria is used [51,52]. This is because bioremediation bacteria are generally resistant to acidic environments. At the same time, one well-known hydrocarbon component of bioremediation products is the simple organic compounds resulting from oxidation reactions in the form of alcohols, aldehydes, ketones and possibly carboxylate group compounds [18,53,54]. Carboxylic compounds from bacterial bioremediation products can change the habitat conditions (media) to a more acidic state. Changes in the medium’s acidity is narrowly between 0.3 to 1.4 [26,50]. This condition is usually accompanied by an increase in temperature so that the degradation activity of bacteria decreases, or the resistance of bacteria becomes weak. If there is an increase in salinity in this condition, especially if accompanied by movement due to currents, it can result in significant bacterial cell death [16,26,51,55]. This is not a significant value, but the nature of bacteria makes them quite sensitive to environmental changes. Therefore, this may cause their performance to drop, or may even kill the bacteria [32,48]. This condition is the limiting factor for the performance of bacterial bioremediation [48,50,56].

In the aquatic environment, especially in marine ecosystems, there are several types of biota, such as sponges, which are known to be often used as objects for the biomonitoring of the pollution of hydrocarbon components and heavy metals; some types of sponges are even used as references or bioindicators in analyzing the levels of PAHs and heavy metal contaminants [54,55]. In addition to biomonitoring and acting as bioindicators of this type of GTP contamination, it is known that this sponge can perform the degradation of hydrocarbon components and the biosorption of heavy metals [7,56,57,58]. This is based on research results that show the ability of sponges to live and thrive in an environment contaminated with GTP pollutants [7,59]. The development of knowledge about the degradation and adsorption function of marine sponges against pollutants was revealed after discovering that these sponges can achieve a mutualistic symbiosis with microorganisms, especially bacteria [60,61,62].

The bacterial–sponge symbiosis model is intensified when the sponge habitat is exposed to hydrocarbon or heavy metal pollutants, or perhaps both, as the sponge tries to survive in the extreme conditions of its growing environment [63,64,65]. At the same time, symbiont bacteria also use these conditions to produce a mucus substance that behaves as an enzyme, which is then spread on the surface of the sponge’s body to avoid the toxic pollutant [66,67,68]. Internal sponges also independently stimulate immunity in themselves against all forms of predators and changes in their habitat by producing metabolic substances [53,69,70].

The types and populations of sponges are vast, so the right sponge selection must be made with symbiont bacteria that have the potential and ability to degrade and adsorb. This can be done by selecting sponges in their habitat, especially those with dark colors or smooth body surfaces because these are coated with mucus [20,21,71,72,73]. Bacterial symbionts from the selected sponge are then isolated to obtain a single isolate [74,75]. The phenotypic analysis of sponge symbiont bacteria through biochemical tests using standard reagents needs to be carried out to ensure that these symbiotic bacteria can perform the functions of PAH degradation and heavy metal adsorption [6,76]. Bacterial symbionts are potentially useful if they react positively with several biochemical reagents, especially Methyl Red, Voges Proskauer, citrates, lactose, catalase, nitrate reduction and indole reagents [77,78]. Genotypic analysis of bacterial symbionts is important to obtain complete information related to bacterial species and strain and the number of DNA base pairs, and it is also possible to carry out genotypic analyses of these symbiotic bacteria using 16S rRNA sequences [36,79,80,81].

The degradation function of sponge symbiont bacteria used for qualitative analysis can be ensured by the identification of several bacterial cells on media containing hydrocarbon components, such as pyrene [82,83,84]. Bacteria that can adapt to an environment containing PAHs are characterized by their activity after being incubated for ±24 h. This situation indicates that bacteria can carry out the biodegradation of polyaromatic hydrocarbon pollutants [85,86]. A preliminary qualitative test can determine the adsorption function of sponge symbiont bacteria by inserting ±1 mL of bacterial suspension into a medium containing heavy metal contaminants after being incubated for ±24 h and then measuring the optical density (OD600) of the interaction medium. If there is an increase in the turbidity or absorption, this indicates that the symbiont bacteria could perform the adsorption function on the heavy metals tested [87,88].

The mechanism of the bioremediation of hydrocarbon components is slightly different between aliphatic and aromatic hydrocarbons [89,90]. The mechanism of the degradation of hydrocarbon components generally proceeds via microorganisms, especially sponge symbiont bacteria, operating through oxidation reactions or biochemical reactions at the molecular level [69,91]. This involves the entry of metabolic substances or enzymes (dioxygenase) produced by the bacteria into the structure of hydrocarbon molecules that act as substrates to enable these molecules to undergo destruction, which causes the molecules to break down and produce organic compound products containing hydroxyl functional groups, and which then turn into keto-enol compounds [92,93,94].

After forming a keto-enol complex as the initial component of the remediation product, the oxidation reaction against PAH contaminants should continue. Further oxidation of these complex molecules produces organic compounds containing aldehyde, carboxylic, and perhaps ketone groups. [16,95,96]. Ideally, the oxidation reaction proceeds until compounds that can enter the metabolic cycle consist of simple organic molecules. [26,97]. The ability of bacterial cells to survive pressure fluctuations in pH, salinity and temperature significantly impacts the persistence of PAH bioremediation [42,43]. The oxidation reaction of hydrocarbon components can proceed if the ideal conditions required for the degradation of bacteria are met [32,48,98]. Generally, the degradation performance of bacteria decreases when the oxidation produces a carboxylic acid product [50,58,99]. Another factor that can inhibit the bacterial degradation of the substrate (reactant) of the hydrocarbon component is the low solubility of the hydrocarbon component, making it difficult for bacteria to penetrate [71,100,101].

The rate of the bacterial biodegradation of hydrocarbon contaminants varies in the range of 35–97%. Several factors cause this, such as the type of bacteria used; the type of hydrocarbon pollutant (aliphatic or PAHs) [26,50,102,103]; the interaction time; the degradation method; the concentration of hydrocarbon components used as reactants; the number of bacterial cells; the presence or absence of nutrition; oxygen injection (aeration); the scale of experiments carried out and other factors [32,48,104,105]. Variations in the level of bacterial degradation of hydrocarbon components indicate that multiple factors influence bioremediation [58,106]. Two things always occur in the biodegradation of hydrocarbon components (subtracts) by bacteria (degraders), namely: (1) The biodegradation of hydrocarbon pollutants by bacteria through an oxidation reaction pathway involving enzymes produced by bacteria in response to the presence of toxic substances in their growth habitat [71,107,108]. (2) A decrease in the performance of bacterial degradation when degradation products are formed as carboxylic compounds. These two things cause the degradation of hydrocarbon components, especially PAHs, to be incomplete, or may prevent the 100% degradation of the substrate [50,109]. Pyrene biodegradation generally stops at the stage at which the benzene component is formed [32,110,111]. Under these conditions, it is assumed that all bacterial cells will have died [110,112]. 

These data make it possible to modify the biodegradation of hydrocarbon components using a consortium of bacteria in diverse species or a consortium of microorganisms (a mix of bacteria and fungi) [51,92,113]. The first modification involves a consortium of certain species of bacteria (X) which have a high degradation activity, is expected to work at the beginning of contact, and is combined with bacteria (Y), which have a slow adaptation rate and continue the degradation process when the cells of the X-species bacteria have died [114]. The second modification, a consortium of X-species bacteria that perform degradation when combined with fungi of type (Z), is more tolerant of acids that can degrade hydrocarbons in acidic media [115,116]. This method is one of the potential alternatives for developing bioremediation technology. It is hoped that all hydrocarbon components will be wholly or 100% degraded and produce the final product of simple organic compounds in the form of salicylic acid and similar, which are environmentally friendly [117,118].

## 2. Polycyclic of Aromatic and Heavy Metals Bioremediation Analysis Instrument

Among the 16 other PAHs, pyrenes are placed in the middle based on their molecular size, their number of rings (four), their toxicity level, and their water solubility [26,71]. Pyrene is considered the most toxic pollutant since it can come from various sources, such as combustion activities and the petroleum processing industry [19]. So far, pyrene has been used as a substrate in bioremediation investigations of PAHs [58,113]. This is why pyrene was selected for this bioremediation study. The bioremediation of global pollutants necessitates applying new technologies and innovations to improve remediation efficiency, such that the natural balance, especially in marine ecosystems, can be maintained [119]. Bacterial consortia in bioremediation represent a new approach and are needed in the future. The analysis of the performance and efficiency of bacterial remediation against PAHs generally uses instruments such as gas chromatography–mass spectrometry (GC-MS) [3,57], Fourier-transform infrared spectroscopy (FTIR), scanning electron microscopy (SEM), energy-dispersive X-ray spectroscopy (EDS) and X-ray diffraction analysis (XRD). In contrast, the remediation of heavy metals by bacteria generally uses atomic absorption spectroscopy (AAS) analytical instruments; inductively coupled plasma (ICP) can be combined with optical emission spectroscopy or mass spectrometry (MS) [120,121,122,123,124]. The combination of analytical instruments in pollutant bioremediation can be useful, especially for the bioremediation of pollutants containing two or more types [122]. 

Waste generated in the petroleum processing industry in the form of sludge, which generally contains hydrocarbon component pollutants, both aliphatic and aromatic, also contains heavy metal toxic materials, so a combination of analytical instruments is often used to obtain data related to performance, efficiency, mechanism and remediation products, including models of and changes to the pollutant material during remediation [125,126]. Researchers often use a combination of analytical instruments in the bioremediation of petroleum sludge waste with a single bacterium/bacterial consortium, such as GC-MS, FTIR and AAS [20,57,121]. SEM, EDS and XRD instruments usually observe changes in the surface shape or structure of bacteria-heavy metal complexes through extracellular bonding [120,121,122,123].

## 3. Bacterial Performance in Pollutant Bioremediation

Bioremediation innovation using a consortium of bacteria in pollutant remediation is a global trend, especially for PAHs and heavy metal pollutants. It aims to improve remediation efficiency and performance such that the final product of remediation is given in the form of simple organic compounds which are environmentally friendly and no longer cause health effects on living things in their environment [51,91,110].

The relationship between single-species biodegrader remediation performance on PAH components and bacterial cell growth based on interaction time is presented in Figure 1.

The bioremediation of PAHs utilizing a single type of microbe (bacteria) is frequently inefficient. These bacteria perform poorly in the breakdown of PAH substrates, both when conducted on a laboratory scale with a single substrate pyrene and when used in natural conditions that are difficult to manage. This is due to the fluctuating influence of the external environment, as well as the salinity and pH levels [48,106,107]. This occurs because the biodegradation of PAHs by bacteria takes place in several stages, generally starting with an oxidation reaction and then destroying the substrate’s molecular structure. The substrate degradation process, via both biostimulation and bioaugmentation methods [16,127,128], using bacteria, continues until transition products are obtained in the form of acidic compounds (carboxylic) [8,9,10,11,26,33,56]. At this stage, the performance of bacteria is often reduced significantly due to their inability to tolerate an acidic medium [28,71,77,113]. As a result of this process, bacterial cells cannot continue the process of cell division, so there is no more prolonged regeneration of bacterial cells that grow to continue the degradation process [58,116,124]. The degradation step involves the formation of carboxylic acid compounds, but most bacteria cannot tolerate acidic conditions, so this step in the bioremediation method is often called the rate-limiting step of degradation [11,23,41,50,116].

The reduction in PAH components (pyrene) by bacillus bacteria (Figure 1) increases with increasing interaction time, followed by an increase in the growth of bacillus cells [38,51]. The growth of bacillus cells appears to cease on the 18th to 20th days of interaction (Figure 1). Under this condition, the growth activity of bacillus cells was considered non-existent, so the biodegradation process of the pyrene substrate also stopped, while the pyrene residue remained at ±40% of the initial amount [30,129].

Figure 1 above illustrates the weaknesses or limitations of using one type of microorganism species (bacteria) in the biodegradation process. This illustration shows the performance of bacillus used in pyrene degradation for 18 days [6,109]. Bacteria is used this time to carry out the degradation of pyrene through the adaptation of the interaction environment, cell growth and multiplication, and stationary and cell–cell death phases [111,113]. Bacterial degradation results in the gradual destruction of the pyrene molecular structure following the growth and development of bacterial cells, resulting in the production of transitional organic compound products until the bacteria reach the degradation stage of the production of carboxylic acid components [44,129]. The concentration of ±40% pyrene remaining at the end of the biodegradation process is still high, and if this enters the environment, the safety of living creatures in the area is not guaranteed [4,98]. 

Bioremediation technology continues to develop today. One of the developments in bioremediation technology is the innovation of using a consortium of bacteria to remediate toxic and carcinogenic PAHs [22,105,124]. The study of the performance and efficiency of biodegradation via the application of consortia or groups of bacteria tolerant to the toxicity of PAHs (hydrocarbonoclastic) [68,113] is presented in Figure 2.

An illustration of biodegradation in aquatic ecosystems by utilizing pyrene as the primary substrate for degradation is shown in Figure 1, and the biodegradation performance using a bacterial consortium is shown in Figure 2. Studies using hydrocarbonoclastic biodegraders in the bioremediation of PAHs can give better results than those using single species of bacteria [81,106]. The application of consortium bacteria in the bioremediation of PAHs is considered to achieve a higher performance and substrate degradation power, and is more efficient, as it is estimated that the bioremediation performance of PAHs increases to 100% (Figure 2) [27,60,126]. The time required by the consortium bacteria to degrade potential PAHs is less than 20 days, which can be inferred from the consortium bacterial cells that still show growth [51,92]. This condition indicates that bacterial cells can still carry out cell division and the degradation of hydrocarbon components for use as an energy source [5,31].

Studies conducted regarding the bioremediation of hydrocarbon components using microorganisms have applied different methods, such as biodegradation, biostimulation, bioaugmentation or a combination of these methods (Table 1) and have shown that none of the experiments succeed in degrading hydrocarbon pollutants with an efficiency reaching 100% [94,128]. The study of heavy metal bioremediation by microorganisms also showed that no single type of bacteria could absorb heavy metal contaminants in waste with 100% efficiency [52,116,130].

The results of this research indicate the need to develop a bioremediation technology for hydrocarbon pollutants (PAHs) and heavy metals. One of the innovations in remediation engineering is the use of several types of bacteria that can bioremediate PAHs and heavy metals to achieve 100% remediation efficiency [37,60,66].

Sea sponges generally undergo a mutualistic symbiosis with microorganisms, especially bacteria [31,113]. Research on the bioremediation of waste containing hydrocarbon components (aliphatic, aromatic) using several types of marine sponge symbiont bacteria shows the ability of these bacteria to degrade hydrocarbon components (Table 2) [12,100]. The search results above show that there are three groups of symbiotic sponge bacteria (*Bacillus, Pseudomonas* and *Acinetobacter*) [38,52,127]. These bacteria showed good biodegradation performance against hydrocarbon components [116]. The research findings related to tracking the remediation performance of sponge symbiont bacteria against pollutants containing hydrocarbon components show that these bacteria are generally isolated from sponges whose body surface is covered with mucus, or dark-colored sponges [73,130]. This has to do with the dynamics shown by sponges suspected of being exposed to pollutants in their habitat, which stimulate themselves to survive in that environment by producing mucus substances [13,20,131].

Similar research has also been conducted to evaluate the capability and efficacy of symbiotic sponge bacteria in removing heavy metal pollution (Table 3). Numerous investigations indicate that certain varieties of bacteria isolated from sea sponges with mucus-coated body surfaces can also adsorb multiple types of heavy metal pollution [32,58,77,132]. In general, symbiotic sponge bacteria are effective and show a great performance against heavy metal adsorption [6,25,133].

The adsorption patterns of sponge symbionts on several types of heavy metal pollutants are different from those of the biodegradation of PAHs, which tends to be directly proportional to the duration of the interaction [6,28,109]. The adsorption of heavy metal pollutants by bacteria occurs extracellularly through the binding of heavy metal ions by extracellular polymers produced by bacterial cells, which act as negatively charged biosorbents on the cell surface that bind and form complexes with positively charged heavy metal ions [59,127]. 

The analysis of the bioremediation pattern of marine sponge symbiont bacteria against several kinds of heavy metals showed that the optimal adsorption of heavy metal pollutants by bacteria generally occurred at a contact duration of 2–6 days or when the bacteria had passed the adaptation phase in a new environment exposed to heavy metal toxins [20,130]. Bacterial remediation activity decreased until it reached a contact time of 20 days. This biosorption pattern is illustrated in Figure 3.

This phenomenon indicates that the adsorption model involves an extracellular ionic bond between the positive pole of the heavy metals and the negative pole of the bacteria on the surface, such that adsorption can occur very quickly when the negative surface of the bacteria is active [60,65,132,134]. The contact duration of 6 days is considered the general period required for the fishery to reach the saturation stage, whereat most of the negative poles of the bacterial surface will have formed ionic bonds with heavy metal ions [35,124]. Under these conditions, the bacterial cells can no longer continue their adsorption activity, and the process towards the division phase and cell growth is declared to have stopped [28,107].

## 4. Process and Mechanism of Pollutant Bioremediation by Marine Sponge Symbiont Bacteria

The degradation process of PAH components by bacteria differs from the heavy metal adsorption process [65,131]. This difference also pertains between the degradation mechanism of PAHs and the mechanism of heavy metal adsorption by bacteria as a bioremediator, despite using the same bacterial species to remove different types of pollutants (PAHs and heavy metals) [16,24,25]. The research on bioremediation using bacteria raises several assumptions that can be scientifically justified [17,42,43].

### 4.1. Processes and Mechanisms of Biodegradation of PAHs

The pyrene biodegradation process using the *Bacillus pumilus* strain GLB197 isolated from marine sponge *Niphates* sp. (Table 2) [124] is illustrated in Figure 4. The illustration shows that the degradation performance when using one type of bacteria is not significant, or the degradation will be incomplete [9,11,31,135].

One of the PAHs (Figure 4a) interacts with the *Bacillus pumilus* (Bp) bacterium (Figure 4b); this contact leads to the Bp bacterial population decomposing PAHs (Figure 4c). In addition, the deterioration that occurs continues to weaken when the Bp population decreases as a result of the increasing acidity of the remediation media. The success of bio-remediation decreases as this technique is implemented in an unregulated, unrestricted environment. For instance, in marine ecosystems, the elements that suppress the performance of deterioration, such as salinity, currents, and temperature, are becoming increasingly variable (Figure 4d). The population of Bp decreases because the remaining bacteria are unable to divide; as a result, the process of degradation barely progresses (Figure 4e), until the point at which all Bp communities have died, and the PAHs remain undigested (Figure 4f) [16,26,110,113,116].

The process of pyrene degradation by bacillus begins with adaptation in the medium, which usually lasts 1–3 days [129]. Then the bacteria enter the enlargement phase, cells divide to form colonies, crowding around the pyrene pieces until the cells enter the pyrene body and cut the pyrene component until it decomposes into small particles (Figure 4) [46,136]. Suppose the bacteria are still able to carry out the remediation activity. In that case, each of these bacterial communities takes pieces to continue the degradation process until the bacterial cell activity stops, perhaps because it has been killed by the changing of the medium to become more acidic [15,34,137].

The process of pyrene degradation by *bacillus* is similar to the bioremediation mechanism that occurs, and the difference is that the bioremediation mechanism takes place at the molecular level or in metabolic substances (micromolecules). In contrast, the remediation process occurs at the component or macromolecular level [11,34,118]. The degradation mechanism of PAHs (pyrene) by bacteria is known as a cycle. Namely, one cycle involves a series of changes in the molecular structure of PAHs from pyrene (four aromatic benzene rings) to phenanthrene (three aromatic rings). The following cycle changes phenanthrene to naphthalene (two aromatic rings; see the imaginary line (Figure 5)), and so on until the conversion of benzene (one aromatic ring) into simple non-aromatic organic compounds that are environmentally friendly [83,117].

The mechanism of pyrene degradation by *Bacillus pumilus* strain GLB197 [113] is similar to the degradation pathway of PAHs by *Cycloclasticus* sp. [110,112,116] and has been adopted into the mechanism of pyrene degradation by *Mycobacterium* sp. [50,91,99,132], combined with genomic analysis and experimental analysis (Figure 5). The mechanism of pyrene degradation is described through an oxidation reaction in four stages of change or one cycle, i.e., starting with the change that produces the cis/trans transition product 4,5-dihydrodiol-pyrene (Figure 5a), which is then converted to 4,5-dihydroxy pyrene (Figure 5b), then to 4,5-dicarboxylic acid phenanthrene (Figure 5c) and finally to the product phenanthrene-4-carboxylate transition (Figure 5d) [17,46,58,64,99].

The fourth stage of the first cycle is the first point of vulnerability for bacteria because, at that stage, a carboxylic acid transition product is formed, which causes the acidity of the media to increase [110,112,113]. Under this condition, the bacterial cells are susceptible, so the potential cell activity decreases drastically, and they can even undergo mass death [91,99,138]. The mechanism of the conversion of pyrene to the transition product of phenanthrene can be called destruction, namely, the destruction of the pyrene molecular structure or the open aromatic ring [117,136]. 

### 4.2. Process and Mechanism of Heavy Metal Bioabsorption

The process of heavy metal adsorption using bacteria is similar to the process of using ethylene diamine tetra acetate (EDTA) and a heavy metal (X) to form an X-EDTA complex [55,136]. The mechanism of heavy metal adsorption involves forming extracellular ionic bonds on the surfaces of bacterial cells that are negatively charged with positively charged heavy metals [16,118]. The process of the adsorption of heavy metal ions lasts for a shorter duration than the degradation of PAHs [25,128]. The adsorption process continues until the saturation point is reached [65,96]. The mechanism and changes in heavy metal adsorption by bacterial cells can be observed using analytical instruments such as SEM, EDS and XRD, while determining the adsorption efficiency can be done using AAS or ICP [125,139,140].

## 5. Parameters of Pollutant Bioremediation

Several changes can be observed either directly by observation or by using analytical instruments to assess the performance of microorganisms (bacteria) in the bioremediation process of PAHs and heavy metal pollutants [141]. These changes are indicators and parameters that can be used to infer the occurrence of pollutant bioremediation activities by bacteria [27,95].

### 5.1. Biodegradation of PAHs

The biodegradation parameters of PAHs, which indicate the presence of degradation activity performed by the bacteria, include the following: (1) The growth of bacterial cells can be seen from the increase in the optical density of the interaction medium (OD600) [5,53]. The increase in OD600 medium is an indicator that bacterial cells have passed the adaptation phase and are heading to the phase of enlargement and division [16,33]. At this stage, the degradation activity of PAHs has taken place. (2) The increase in the acidic properties of the media is a manifestation of the work of bacterial cell degradation, which has succeeded in destroying the molecular structure of PAHs, forming several new components, among which are carboxylic compounds, which results in increased media acidity [110,112]. This stage is a vulnerable period for bacteria, which can result in cells not being able to enlarge and divide, and the bacterial cells may even be threatened with mass death [103,134]. (3) The increase in the temperature of the interaction media due to the formation of new compounds resulting from degradation. This parameter only increases by a few points, generally in the 0.4–1.2 °C range [23,37]. (4) The emergence of gas bubbles indicates that these bacteria are part of an aerobic group that requires oxygen in carrying out PAH remediation. The oxygen demand of the interaction media is met by aeration using a shaker, or it can be injected directly (5). The smell of fermentation is a characteristic feature of the enzymatic reaction that occurs at the stage of the destruction of the molecular structure of PAHs [17,46,68]. The existing enzymes are produced by bacteria as a response of the cells in order to defend themselves in extreme environments exposed to PAHs [66,67]. (6) New peaks are identified on the GC-MS chromatogram [120,121]. The peaks recorded with varying abundance are valid evidence of all the previously described events (points 1–5) [8,126]. The detection of functional groups of organic compounds via the FTIR chromatogram strengthens the GC-MS data showing that the degradation products produce organic compounds, one of which is a carboxylic component containing carbonyl and hydroxyl groups [122,142,143].

### 5.2. Heavy Metal Bioadsorption

The parameters of heavy metal adsorption by bacteria that can be observed and measured include increased media turbidity, a limited change in the pH ranging 0.2–0.4, temperature increases in the narrow range of 0.3–0.8 °C, gas bubbles appearing, and the smell of fermentation being very weak [3,50]. Changes during the adsorption process are not strong enough to be described in detail, as in the biodegradation of PAHs [87,106]. This is because the remediation occurs in the form of adsorption, with the extracellular ionic bonding of the negative part of the bacterial surface to the positive charge of the heavy metals [7,37,61]. Therefore, at a certain duration of contact, a saturation point can be reached in the media, but this saturation point cannot be observed directly. The saturation point is inferred if adsorption efficiency is determined by bidaily contacts using the AAS instrument [20,60]. In this case, the saturation point is assumed to arise in the contact period of 6–20 days (Figure 3).

## 6. Development and Formulation of Remediator Bacteria Consortium

The performance and efficiency of PAH pollutant biodegradation can be improved by making experimental modifications, especially using consortium bacteria. Similarly, the capacity and level of bacterial adsorption to heavy metal contaminants can be improved with the modification of a consortium of bacterial biodegraders [35,92,130].

### 6.1. Hydrocarbonoclastic Bacteria

The application of hydrocarbonoclastic bacteria (R-1) in the biodegradation of PAHs is one of the modifications that is considered to improve the performance and efficiency of bioremediation [34,95]. Several types of bacteria can be used, for example, *Bacillus pumilus*, *Pseudomonas stutzeri* and *Acinetobacter calcoaceticus* [9,127], all of which can biodegrade PAHs (hydrocarbonoclastic bacteria) [99,144]. These bacteria were formulated as a consortium bacterial suspension and interacted with pollutant PAHs, e.g., pyrene (Figure 6) [112,145].

This modification is believed to increase the degradability of PAHs because each type of bacteria can enact degradation in parallel, enabling them to complete one cycle of the conversion of pyrene to phenanthrene in a shorter time [71,90]. The study of the potential of the R-1 consortium bacteria was intended to help remediate waste containing hydrocarbon components, especially PAHs, with high performance and efficiency [38,145]. 

The bioremediation of PAHs (pyrene) utilizes a consortium of bacteria (hydrocarbonoclastic), namely, *Bacillus pumilus* (Bp), *Pseudomonas stutzeri* (Ps), and *Acinetobacter calcoacceticus* (Ac) (Figure 6a,b). The bioremediation process appears to proceed more rapidly because the bacterial population is large, and the three types of bacteria (Figure 6c) undergo distinct remediation procedures based on their features (Figure 6d). This also indicates that bioremediation is more effective than the use of a single bacterium (Figure 6e) (Figure 4). This causes the pyrene to be significantly destroyed (Figure 6e). However, it is expected that consortium bacteria cannot completely digest pyrene since the state of the bacterial cells is very poor due to the increasing acidity of the medium caused by the creation of degradation products of aldehyde and carboxylate components. Other hypotheses attribute the inability of bacterial cells to divide to a lack of energy and nutrients or the toxicity of PAHs on bacterial cells. [24,26,116]. 

The degradability of bacteria was significantly increased, resulting in a high-efficiency degradation [50,110]. Thus, all pyrene could be degraded to produce the final product in the form of simple non-aromatic organic compounds that are environmentally friendly (Figure 6) [44,117]. The problem of the limiting factors in the PAHs degradation process in relation to the formation of transition products of carboxylic acid compounds is also assumed to be minimized, so that bacterial cells can work continuously to convert carbon elements into energy [82,110].

### 6.2. Metalloclastic Bacteria

We have considered the performance, capacity and efficiency of heavy metal adsorption using several types of bacteria in the form of a consortium, or bacteria of the metalloclastic category (R-2) [9,131]. An increase in the capacity and efficiency of heavy metal pollutant adsorption occurs due to the abundance of negative poles on the bacterial cells’ surface, allowing the formation of ionic bonds that occur at the surface at almost the same time [28,77].

In general, bioremediation of PAH and heavy metal-polluted soil is simpler than in aquatic ecosystems [32]. This is due to the relatively static soil media, which enables superior bacterial bioremediation performance since it suits the features of bacteria as a decomposer of the PAH molecular structure and turns it into energy to be used for subsequent bacterial growth and regeneration. Due to the adhesive nature of bacteria, heavy metal pollution can become entrapped in their bodies.

The effectiveness of bacterial bioremediation in aquatic ecosystems against PAH and heavy metal pollution appears to be more dynamic and weak. This assumption is brought about by external influences (salinity, pH, temperature, currents or waves, and mineral availability) [26]. Another aspect is internal bacteria (tolerance level, metabolic processes, cell resistance) [106] Salinity, pH, and temperature are thought to be the most influential factors impacting the performance of bacterial biodegradation of PAH and heavy metal contaminants, whilst all other factors are assumed to be constant. The condition of bacteria as bioremediators in extreme environments contaminated with PAHs and heavy metals has been under strain due to salinity and the toxicity of pollutants since the beginning of the interaction. In contrast, these bacteria are still able to perform remediation duties. When a new material in the form of a carboxylic acid component is created as a result of remediation, the acidity of the remediation medium and the temperature increase within a restricted range [13,90,107]. This weakens the resilience of bacterial cells and can be lethal if there is an increase in salinity and the presence of currents [48,116].

It is proposed to utilize a consortium of bacteria with diverse metabolic methods to boost cells’ tolerance and resistance, hence improving the performance of pollution bioremediation. However, salinity pressure, pH variations, temperature rises, and mass movement related to tides remain the factors that influence and might diminish bioremediation performance.

The differences in adsorption efficiency due to differences in the reactivity and affinity of each heavy metal, as well as the presence of several types of bacterial cells in the consortium formulation, allow the barriers that suppress the abundance of ionic bond formation to be overcome, because each type of bacteria has a specific adsorption model [6,20]. The positive charge of each heavy metal also affects the bonds formed. The adaptability factor of bacterial cells to an environment containing heavy metals greatly determines the adsorption process. In contrast, the external influence of adsorption, such as the provision of nutrients, and the presence of aeration, do not positively impact the number of ionic bonds formed [35,113]. The consortium of bacteria coded R-2 has high adsorption power and efficiency, and they are intended to be applied in the remediation of waste exposed to heavy metals [77,131].

The bioremediation of heavy metals by bacteria resembles a chelate formation reaction; it involves the adsorption of positive ions from heavy metal contaminants on the surface of bacteria [9,10]. Suppose, according to the ionic size, that a bond is formed that eventually forms a heavy metal complex with the active side of the bacteria. In that case, the ionic size determines the formation of this complex. Heavy metal bioremediation is based on ion trapping. In contrast, PAH bioremediation is based on the breakdown of molecular structures [15,58].

### 6.3. Metallo-Hydrocarbonoclastic Bacteria

In addition to containing aliphatic and aromatic hydrocarbon components, sludge waste originates from petroleum processing and contains several types of heavy metals [65,96]. This condition requires using bacteria that have a biodegradation function and an adsorption function [60]. Several studies have shown that several types of sponge symbiont bacteria can degrade PAH components and adsorb heavy metals, although these ability tests were carried out separately [6,45,146]. Groups of bacteria with multiple abilities are called metallo-hydrocarbonoclastic (R-3) bacteria.

Several types of research have been conducted; it was found that several sponge symbiont bacteria can be included in the group of bacteria coded R-3, these representing a collection of bacteria that can biodegrade PAHs and also have the ability to adsorb heavy metals [34,109].

This study concludes with recommendations and suggestions for developing bioremediation technology for GTP, especially with PAHs and heavy metals that are important to producing and improving remediation efficiency [19,110]. The bacterial consortium of marine sponge symbionts, both hydrocarbonoclastic (R-1) and metalloclastic (R-2), or metallo-hydrocarbonoclastic (R-3), bacteria, have the potential to increase remediation efficiency in various types of waste [51,81,92]. Screening is important to find and categorize bacteria (R-1; R-2; R-3) with different abilities in terms of GTP remediation [1,7,19]. The bacterial formulation codes R-1, R-2, and R-3 can also be developed in the future into crystalline preparations so that these bacteria will be more easily mobilized for rapid culturing at sites exposed to GTP [2,7,147]. The search for marine sponge symbiont bacteria for bioremediation with high performance and efficiency can be traced only to marine sponges whose body surface is covered with mucus, or which is dark in color [59,88].

## Figures and Tables

**Figure 1 biology-12-00086-f001:**
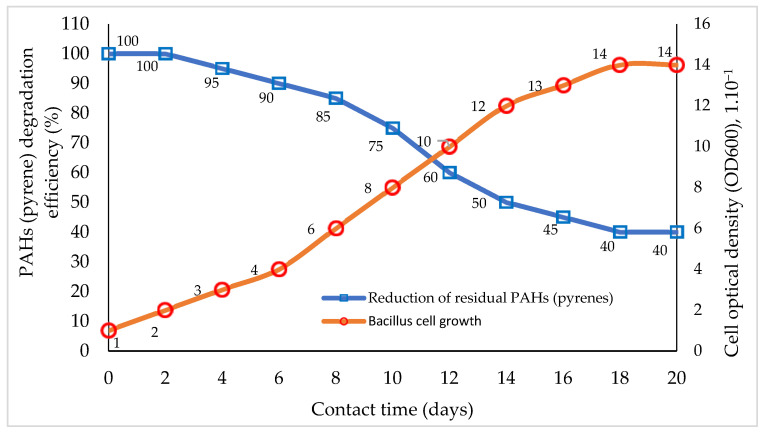
Comparison of the percentage of PAH (pyrene) bioremediation with bacterial cell growth rates based on interaction time. Graphically modified figure [13,16,110,113,116].

**Figure 2 biology-12-00086-f002:**
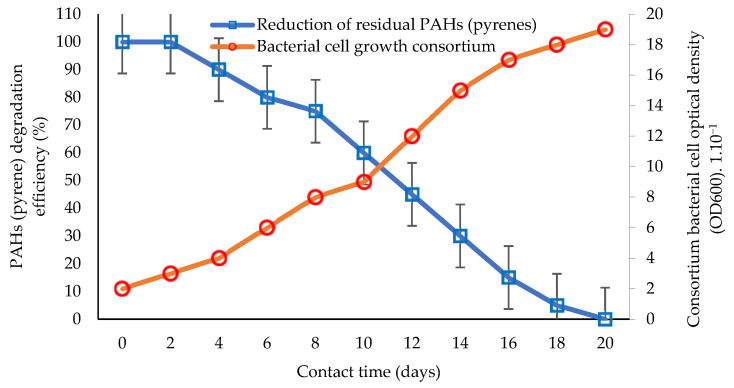
Comparison of the percentage of PAH (pyrene) bioremediation with the bacterial cell growth rate of the consortium based on interaction time. Graphically modified figure [51,64,71,113,116].

**Figure 3 biology-12-00086-f003:**
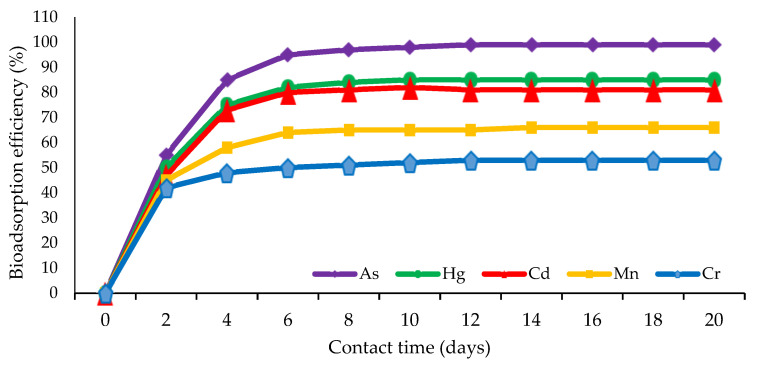
The pattern of adsorption of sponge symbiont bacteria on some heavy metal pollutants.

**Figure 4 biology-12-00086-f004:**
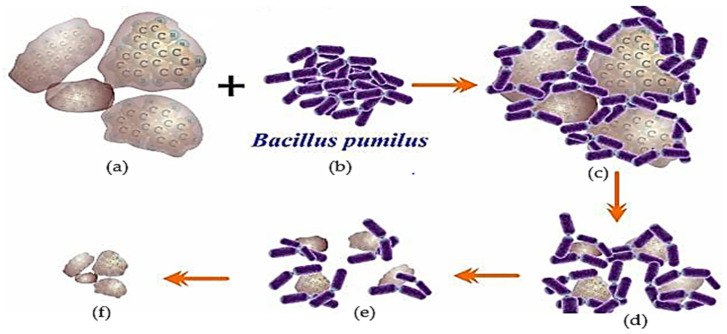
Description of the biodegradation process of PAH (pyrene) contaminants using one type of bacteria, *Bacillus pumilus* type. (**a**) Pyrene as a substrate; (**b**) *Bacillus pumilus* (Bp); (**c**) pyrene-Bp complex; (**d**) Degraded pyrene; (**e**) The population of Bp was significantly reduced and the pyrene concentration was not significantly reduced and (**f**) the remaining pyrene was not degraded, an illustration [16,71,99,116].

**Figure 5 biology-12-00086-f005:**
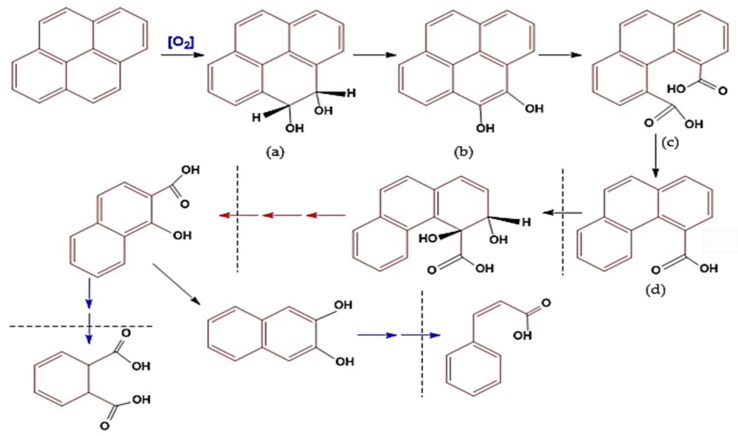
Illustration of the mechanism of biodegradation of pyrene components using *Bacillus* sp. bacteria, through the following metabolic reactions: (**a**) Pyrene molecules undergo oxidation; (**b**) a keto-enol molecule is formed; (**c**) One benzene ring is broken to form a carboxylic acid compound; (**d**) a simple carboxylate molecule separates to form penanthroic acid. The next degradation process is through an oxidation reaction which leaves benzoate compounds. Figure has been modified based on some references [99,110,113,116].

**Figure 6 biology-12-00086-f006:**
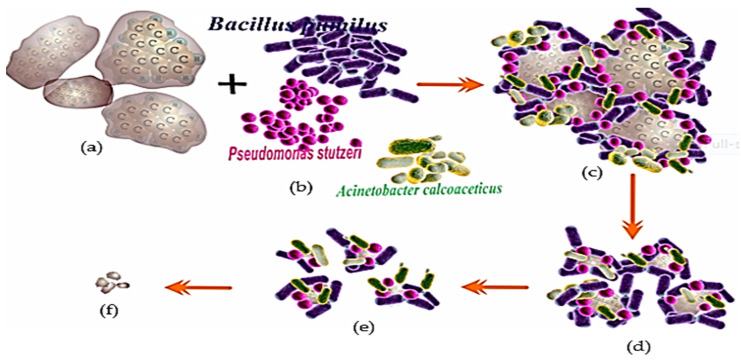
The process of the biodegradation of PAH contaminants uses a consortium of hydrocarbonoclastic bacteria. An illustration, bacterial consortium biodegradation of pyrene: (**a**) Pyrene substrate; (**b**) consortium of bacterial degraders (hydrocarbonoclastic); (**c**) Complex of consortium bacterial-pyrene; (**d**) Pyrene begins to degraded; (**e**) Pyrene is significantly degraded and (**f**) A small portion of pyrene is not degraded and the bacterial consortium population has died [58,99,110,113,116].

**Table 1 biology-12-00086-t001:** Results of recent studies on the application, performance and efficiency of bacteria in the remediation of polycyclic aromatic hydrocarbons (PAHs) and heavy metal pollutants.

Type of Contaminant	Bioremediation Method	Test System	Interaction Duration	Removal Efficiency	Conclusion	References
Pyrene (±10 mg/kg)	Biosurfactants (Biodegradation)	Soil microorganism	10 days	±60%	The biodegradation process can occur due to the ability of rhamnolipids to convert carbon into energy sources	[111]
Phenanthrene (±1.0 mg/L)	Biodegradation	Using soil adsorption reactor	more than 50 days	90.0%	No significant effect of the observed biodegradation efficiency of surfactants	[54]
PAHs (±574 mg/kg)	Biodegradation	Soil microorganism	84 days	72.0–77.0%	The formation of surfactants marks the ongoing biodegradation process	[74]
Pyrene (±100 mg/L)	Biodegradation	*Sphingobacterium* sp. strain 21	30 days	±38.3%	The biodegradation performance of pyrene increases at the contact period of 6–20 days	[71]
Pyrene, phenanthrene and others (±6 mg/kg)	Biosurfactants (Biodegradation)	Soil microorganism	±35 days	58.7%	Biosurfactants *(rhamnolipids)* can only carry out biodegradation until the 7th day, then the PAHs biodegradation process does not appear until the 35th day	[85]
Phenanthrene (±1.0 mg/L)	Biodegradation	Flake model	14 days	60.0%	Using *Rhamnolipids* as surfactants can increase the efficiency of biodegradation at a concentration of 100 mg/L	[127]
PAHs (±1.5 mg/g)	Biostimulation	Soil microorganism	56 days	±99.0%	The biostimulant effect can increase biodegradation kinetics	[94]
Pyrene (100 mg/L)	Biodegradation using vial reactor	*Alcaligenes faecalis* strain Cu4-1	25 days	97.7%	The products of pyrene biodegradation via the two types of bacteria are relatively different, indicating that there are different metabolic pathways that are influenced by these types of bacteria	[64]
		*Bacillus Cereus* strain MER-8		93.2%		
Petroleum refinery waste (±144 g/kg)	Combined biostimulation and bioaugmentation	*Microorganisms in vial*	120 days	57–75%	Modification of the method by applying a combination of biostimulation and bioaugmentation to increase remediation efficiency	[128]
Alkanes (initial concentration not determined)	Bioaugmentation	activated microcosm consortium	85 days	35–66%	The use of the adapted microcosm consortium can degrade the hydrocarbon component as a substrate to produce biosurfactants	[95]
Pb(II) and Cd(II)	Bioadsorption	*Burkholderia fungorum FM-2*	7 days	50 mg/L and 400 mg/L	*Bacillus fungorum* strain FM-2 is tolerant to PB(II) and Cd(II) or can carry out the function of bioadsorption of heavy metals	[129]
Pollutant Cd and Hg	Bioadsorption	*Pseudomonas* sp., *Salinobacter* sp., *Streptomyces* sp., *Roseobacter* sp., *Vibrio* sp., *Sac-charomonospo-ra* sp. and others isolated from marine sponge *Fasciospongia cavernosa*	7 days	Preliminary test	This sponge symbiotic bacteria is able to survive in habitats contaminated with heavy metals mercury and cadmium	[130]
Pb(II) and Cd(II)	Remediation in oxidation method	Natural adsorbent available in aquatic habitat	2 days	9.0 mg/g and 8.9 mg/g	Natural adsorbent found in the aquatic environment in remediating Pb(II) and Cd(II) pollutants. The isotherm data were processed using the Langmuir approach, showing that lead remediation is endothermic and cadmium is exothermic	[28]
Ions Co, Pb, Cu, Zn	Bioadsorption	*Rumex crispus L*	7 days	83.5–91.0%	The findings reveal that the heavy metal absorption mechanism occurs on the surface of the biosorbent to form a metal–biosorbent complex	[60]

**Table 2 biology-12-00086-t002:** Various studies on the biodegradation performance of sponge symbiont bacteria on hydrocarbon components.

Types of Hydrocarbon Contaminants	Sponge Symbiont Bacterial Species	Type of Sea Sponge	Interaction Duration	Removal Efficiency	Conclusion	References
Pyrene (±100 mg/L)	*Bacillus licheniformis* strain ATCC 9789 (Bl)	*Auletta* sp.	30 days	±39.0%	The performance and biodegradation kinetics increased during the contact period of 10–25 days, then slowed down to day 30	[71]
PAHs (Anthracene and pyrene)	*Bacillus pumilus* strain GLB197	*Niphates* sp.	25 days	Antracene (21.9%) Pyrene (7.7%)	The consortium of three types of bacteria isolated from sea sponges can carry out the function of biodegradation of pyrene and anthracene components, but the performance is less significant, presumably due to competition for carbon as an energy source	[113]
	*Pseudomonas* *stutzeri* strain SLG510A3-8	*Hyrtios erectus*				
	*Acinetobacter calcoaceticus* strain SLCDA 976	*Clathria (Thalysias) reinwardtii*				
PAHs	*Pseudomonas* sp. strain Hi1	*Auleeta sp*	Preliminary test on PAHs contaminated media	Observation (qualitative)	All types of sponge symbiont bacteria showed activity on media exposed to PAHs	[78]
	*Bacillus subtilis* strain BAB-1684	*Clathria reinwardti*				
	*Pseudomonas stutzeri* strain RCH2	*Callyspongia sp*				
	*Bacillus flexus* strain PHCD-20	*Hyrtios erectus*				
Naphthalene	*Bacillus* sp.	*Neopetrosia* sp	25 days	±51.4%	Both types of spongy symbiont bacteria can degrade naphthalene, characterized by several parameters, namely, the increased acidity of the interaction medium, increased optical density (OD600), smells of fermentation and gas bubbles are formed	[58]
	*Acinetobacter* *Calcoaceticus*	*Callyspongia* *Aerizusa*		±37.3%		
Pyrene	Sp AB1 and Sp AB2	*Hyrtios erectus*(Sp A)	Preliminary test on pyrene contaminated media	The activity of the two isolates is weak	The activity of isolates against pyrene generally came from sponges whose body surface was covered with mucus. This mucus is thought to have an enzyme character	[76]
	Sp BB1 and Sp BB2	*Clathria (Thalysias)* *reinwardti* (Sp B)		Both isolates did not show activity		
	Sp CB1 and Sp CB2	*Niphates* sp. (Sp C)		Both isolates showed strong activity		
	Sp DB1 and Sp DB2	*Callyspongia* sp. (Sp D)		Both isolates showed moderate activity		
PAHs Naphthalene and Anthracene	Isolate Sp6. B2	*Auletta* sp.	20 days	There is biodegradation activity	The biodegradation activity of Sp6.B2 isolates against naphthalene and anthracene appeared to be more dominant than Sp8.B1 isolates.	[124]
	Isolate Sp8. B1	*Callyspongia* *Aerizusa*				
Aliphatic Components	*Bacillus cohnii* strain DSM 6307	*Niphates* sp.	25 days	Average 48.1%	GC-MS and FTIR detected new organic compounds of alcohol, aldehyde and carboxylic acid groups	[31]
	*Bacillus pumilus* strain GLB197					
Petroleum sludge	*BacillusFlexus* strain PHCDB20.	*Callyspongia* sp.	35 days	Identified 18 types of aliphatic comp. and 2 aromatic comps.	All hydrocarbon components in the degraded sludge are characterized by a decrease in abundance	[13]

**Table 3 biology-12-00086-t003:** Various studies on the bio-adsorption performance of sponge symbiont bacteria against heavy metal contaminants.

Types of Hyd-rocarbon Contaminants	Sponge Symbiont Bacterial Species	Type of Sea Sponge	Interaction Duration	Removal Efficiency	Conclusion	References
Chromium (VI) Manganese (VII)	*Acinetobacter calcoaceticus* strain PHCDB14	*Callyspongia* *aerizusa*	15 days	±63.2%	Both types of pollutants are absorbed to the maximum at a contact period of 3 days	[65]
				±66.8%		
Cr, Zn, Cu, Fe, Co, Mn, Ag and Cd	*Bacillus cohnii* strains DSM 6307	*Niphates* sp.	16 days	Heavy metal pollutant removal efficiency varies	All types of heavy metals tested can be absorbed by the symbiont bacteria isolates *Niphates* sp. and *Clathria (Thalysias) reinwardtii* with varying biosorption performances. Optimum biosorption occurs at a contact period of 4 days	[77]
	*Pseudomonas stutzeri* RCH2	*Clathria (Thalysias) reinwardti*				
Cd^2+^ and As^3+^	Isolate Sp6. B2	*Auletta* sp.	20 days	83.2%, and 82.2%	Optimum biosorption occurred at a contact duration of 5 days, then weakened until the 20th day of the contact period	[124]
	Isolate Sp8. B1	*Callyspongia* *Aerizusa*		99.9%, and 99.9%		
Cr(VI) and Cd(II)	*Bacillus pumilus* strain GLB197	*Niphates* sp.	15 days	56.3% and 61.2%	The optimum bioadsorption of these two types of sponge symbiont bacteria against the two types of heavy metal pollutants tested occurred in the range of 3–6 days of contact	[32]
	*Pseudomonas stutzeri* strain SLG510A3-8	*Hyrtios erectus*		52.7% and 57.8%.		
As^3+^ and Hg^2+^	*Bacillus* *licheniformis* strain ATCC 9789	*Auletta* sp.	16 days	99.9%, and 88.5%,	Biosorption takes place optimally at a contact duration of 3–6 days. Another supporting indicator is the increase in optical density (OD600) and gas bubbles detected in the interaction medium	[131]

## Data Availability

No specific statement regarding data availability.

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
