# Peer review of "Potential Utilization of Bacterial Consortium of Symbionts Marine Sponges in Removing Polyaromatic Hydrocarbons and Heavy Metals, Review"

_biology, 2023, doi:10.3390/biology12010086_

Round 1
Reviewer 1 Report
The Title must be modified.. Putting this word “Pollutants Global Trends, A” in the title is obscure. This can be replaced by “PAHs and heavy metals”, which are the focus of this review paper.
L31- “This study” must be replaced by “This review paper” . After all this is a review paper and a synthesis of several papers not conducted by the author themselves
The manuscript is poorly written and need to be edited for English language by a native speaker.
L198: The term Global Trends Pollutant must be corrected here.
Figure 1. Are the data in Figure 1 based on the study of the authors or based on other studies? If based on other studies, pls provide citation. I have same question for Figure 2.
Figure 1. Italicize “Bacillus” in the label
Table 1. Table 1 label . L282 “Results of a recent study” must be “Results of recent studies”
Overal,l I suggest that the paper must be reviewed first by a native English speaker and resubmitted.. The bad use of English has compromised a lot of meaning in the text.. The language usage is also a factor why the manuscript is difficult to read and to follow.
Author Response
Dear Reviewer-1
Thank you for the corrections that have been made, as the purpose of this article is to improve its quality. We have revised everything following the Reviewers' comments and suggestions; nonetheless, it is quite unlikely that our improvements will fulfil the Reviewers' expectations. This is an open-access publication, so it is crucial to write high-quality articles in accurate English.
Thank you very much
Best regards
Ismail Mz and friends

Reviewer 2 Report
I congratulate the authors on their research, it is well structured and robust. I just believe that the title and the introduction carry the wrong message to the reader. From the text it seems that it will deal with several global (organic) pollutants. The text focuses more on PAHs, especially pyrene. So please, I ask the authors to adjust the text to give the correct message of the article. Why the focus on pyrene when there are several PAHs? What about chrysene?
Author Response
Dear Reviewer-2
Thank you for the corrections that have been made, as the purpose of this article is to improve its quality. This is an open-access publication, so it is crucial to write high-quality articles in accurate English. Hopefully, the revisions that have been made can improve the quality of the manuscript, even though it may not meet the expectations of the reviewers. We really hope that reviewers can approve this manuscript, so that it can be published by the journal Biology-MDPI.
Thank you very much
Best regards
Ismail Mz and friends

Reviewer 3 Report
Dear authors:
In my opinion, you have done some interesting work in the important field of bioremediation of marine ecosystems. However, I have made some suggestions that may help to improve your work or make it clearer. The suggested points for improvement are as follows:
Line 28 in “Global trending pollutant” (GTP)
Line 100-101 “Carboxylic compounds of bacterial bioremediation products can change the conditions of bacterial habitat (media) in an acidic environment so that the degradation activity of bacterial cells decreases or may die in bulk. This condition is called the limiting factor for the performance of bacterial bioremediation”.
How much could a pH fluctuate with increasing carboxylic compounds?
Line 164-165, Line 179-181, Line 243-244, Line 247-260, Line 392-393.
In all these sections, it is mentioned that the degradation activity of the hydrocarbon is reduced due to the formation of carboxylic compounds that acidify the medium and that many bacteria cannot tolerate these conditions. Some bibliographical references (23, 41, 111, 112, 51) do not argue this point. The authors should explain and clarify this idea because of its importance in the degradation of hydrocarbons, using appropriate references to the decrease in degradation due to the accumulation of these carboxylic compounds. They should also specify which ecosystems they are referring to.
Line 456 “ 6. Development and Formulation of Remediator Bacteria Consortium”
This section should indicate the effectiveness of a symbiont sponge in its natural unmodified state, to be compared when bioaugmented with metallosclastic and hydrocarbonclastic bacteria.
It would also be interesting to discuss these questions: have these bioaugmentation experiments been done only in the laboratory?; can they be done in situ?; are there any limitations to these practices with respect to another ecosystem?
More information about:
1- It is assumed that the bioremediation study is for marine ecosystems. Frequently in the text and bibliographic references, the two ecosystems of soil and water are mixed. This is the case in Table I where bioremediation methods are mentioned. My suggestion is that the authors make a brief description and/or argue differences and similarities of bioremediation methods in these two habitats, in order to introduce the presence of the two ecosystems in this publication.
2- It would be interesting if the authors could discuss in more detail the bioremediation of heavy metals by the biosorption process: could they comment on why only surface adsorption is involved or can there be other metabolic processes; is the role of the symbiont sponges to immobilise heavy metals or can they also be removed?
Fig1-2.
Authors must indicate the origin of the data shown in the figures (author and references).
Pyrene must be put on the Y-axis, it is the only PAH that is analysed.
It is also necessary to indicate if the values have been obtained in a soil or aquatic habitat.
Fig. 4 and fig.6
The first image of the figure should be labelled.
Fig. 5
If this figure is a modification of an author, the author and the author's reference must be included.
Author Response
Dear Reviewer-3
Thank you for the corrections that have been made, as the purpose of this article is to improve its quality. The author appreciates all corrections and suggestions for improvements from the Reviewers, and we have made revisions despite the possibility that they do not meet the Reviewers' expectations. Hopefully with the revisions that have been made can improve the quality of this manuscript. This is an open-access publication, therefore it is crucial that articles are written in perfect English.
We, the authors, really hope that our manuscript can be approved by reviewers, and can be published by the journal Biology.
Thank you very much.
Best regards
Ismail Marzuki and friends

Round 2
Reviewer 1 Report
This paper has several language flaws that makes it difficult to understand. I do not believe that this was sent for language editing already. Thus, I suggest submitting this one for extensive language editing..
Author Response
Dear Reviewer 1
Thank you for your feedback regarding the linguistic quality of our manuscripts. We also regret for this issue, which prevented reviewers from comprehending this work fully. We indicate that the updated round 1 article has not yet been proofread because, at that time, we had merely submitted it to the local editor suggested by the Biology journal. Here, we present the proofread manuscript as well as the second round of edits, particularly the blue-highlighted passages. Manuscript attached in pdf.
With the second round of editing and proofreading, we will hopefully be able to improve the quality of this article and its language. We sincerely hope that reviewers can understand the importance of this study and agree to its publication in the Biology journal-MDPI.
Note: The second revision of this manuscript uses a proofread manuscript.
Thank you very much
Ismail Marzuki and Friends
October 16, 2022
Reviewer 3 Report
Dear authors:
I would like to explain my decision to "reject" your manuscript. It is mainly based on my disagreement with the hydrocarbon degradation line of argument, since, in this latest version you have submitted, you insist again that the decrease in bacterial activity is due to the production of carboxylic compounds generated during hydrocarbon degradation, i.e. it is due to the acidification of the medium: "Carboxylic compounds in bacterial bioremediation products can change the conditions of the bacterial habitat (media) into an acidic environment, so that the degradation activity of bacterial cells decreases or may die en bloc [16,26,51,55]." This idea is repeated insistently throughout the document as the basis for hydrocarbon degradation, for example in lines 119-123; 185-186; 265-273; 407-409. This idea may be interesting, but it must necessarily be properly justified.
However, I have carefully read the references you have provided to support this argument, such as 16, 26, 51, 48, 112, but unfortunately I have not found any justification for this idea. Only your own articles (references 32 and 113) mention this idea; however, they do not really demonstrate that it is the acidity of the medium that causes the decrease in bacterial growth, and not some other component or cause.
In a marine environment, this explanation would need a more convincing demonstration, as you provide quite a few references related to the degradation of hydrocarbons in soil, not in a marine environment, whose conditions are logically very different. Therefore, I consider that a reasoning developed in a "soil" environment cannot be used to explain what happens in a marine environment (with high concentrations of salts that obviously influence bacterial composition).
After revisions, this explanation, which is key to the manuscript, or a reasonable alternative, has not been considered, and no alternative has been offered by the authors.
Of course, I may be misunderstanding the main idea of your manuscript, but I find no alternatives to the explanation you offer, which definitely hinders my understanding of the manuscript and, in my opinion, prevents the development of its main idea.
This is the reason why I consider that I should not evaluate your manuscript any further, as I have already told the editors.
In any case, I wish you all the best in the process of revising your manuscript, and I hope that my comments will be helpful to you in some way.
Best regards.
Author Response
Dear Reviewer 3
Thank you for the comments and adjustments made by reviewers. In general, we agree with the reviewer's correction, as it aims to improve the quality of the article. It is crucial that the reviewer has pointed out and instructed the author to enter the work's substance. With the second edit, we will perhaps get closer to meeting the reviewers' expectations. Particular revisions have been made to some of the text's statements, including the line:
Details are attached in the pdf.
Note: The second revision of this manuscript uses a proofreaded manuscript.
Thank you
Sincerely From Us
Ismail Marzuki and friends
16 October, 2022
